# Fusion Protein of RBP and Albumin Domain III Reduces Lung Fibrosis by Inactivating Lung Stellate Cells

**DOI:** 10.3390/biomedicines11072007

**Published:** 2023-07-16

**Authors:** Jaeho Choi, Yuna Son, Ji Wook Moon, Dae Won Park, Young-Sik Kim, Junseo Oh

**Affiliations:** 1Department of Anatomy, College of Medicine, Korea University, Seoul 02841, Republic of Korea; hear676@naver.com (J.C.); sonyuna90@gmail.com (Y.S.); mjw6132@korea.ac.kr (J.W.M.); 2Division of Infectious Diseases, Department of Internal Medicine, Korea University Ansan Hospital, Ansan 15355, Republic of Korea; pugae1@korea.ac.kr; 3Department of Pathology, Korea University Ansan Hospital, Ansan 15355, Republic of Korea; apysk@korea.ac.kr

**Keywords:** lung fibrosis, stellate cells, R-III

## Abstract

Activated stellate cells play a role in fibrosis development in the liver, pancreas, and kidneys. The fusion protein R-III, which consists of retinol-binding protein and albumin domain III, has been demonstrated to attenuate liver and renal fibrosis by suppressing stellate cell activation. In this study, we investigated the efficacy of R-III against bleomycin-induced lung fibrosis in mice. R-III reduced lung fibrosis and primarily localized in autofluorescent cells in the lung tissue. Furthermore, we isolated lung stellate cells (LSCs) from rat lungs using the isolation protocol employed for hepatic stellate cells (HSCs). LSCs shared many characteristics with HSCs, including the presence of vitamin A-containing lipid droplets and the expression of alpha-smooth muscle actin and collagen type I, markers for activated HSCs/myofibroblasts. LSCs spontaneously transdifferentiated into myofibroblasts in in vitro culture, which was inhibited by R-III. These findings suggest that R-III may reduce lung fibrosis by inactivating LSCs and could be a promising treatment for extrahepatic fibrosis.

## 1. Introduction

Idiopathic pulmonary fibrosis (IPF) is a chronic lung disease characterized by lung fibrosis of unknown origin [1]. Myofibroblasts, the primary extracellular matrix (ECM)-secreting cells, accumulate and excessively deposit ECM proteins, leading to the destruction of normal alveolar architecture and disrupting gas exchange. Multiple sources of myofibroblasts have been proposed, including resident fibroblasts, epithelial to mesenchymal transition (EMT), and fibrocytes [2]. Recent studies using genetic lineage mapping suggest that pericytes and resident fibroblasts are the major sources of scar-producing myofibroblasts following lung injury induced by bleomycin [3]. IPF leads to death within 2–5 years of diagnosis, and although there has been significant progress in treating IPF, there is still no cure for the disease [4].

Tissue fibrosis can occur in various tissues, including the lungs. Among these, liver fibrosis draws more attention due to its clinical and pharmaceutical aspects, given the frequency of its occurrence. The prevailing consensus is that myofibroblasts observed in fibrotic liver tissue are primarily derived from hepatic stellate cells (HSCs), with a lesser contribution from portal fibroblasts [5]. HSCs, also called liver pericytes, reside within the perisinusoidal space of the liver, comprising roughly 5–8% of all liver cells [6]. These cells, while in a resting state in the healthy liver, demonstrate non-proliferative properties and serve as storage for nearly 80% of the body’s vitamin A (retinol) in the form of retinyl esters enclosed within lipid droplets found in their cytoplasm. In response to fibrogenic stimuli, quiescent HSCs undergo functional and phenotypic changes, referred to as “activation”, and transform into myofibroblast-like phenotype [7]. During this process, distinct features emerge, including the disappearance of cytoplasmic lipid droplets containing vitamin A, which exhibit fast-fading autofluorescence within the range of 330–360 nm. Furthermore, there is a notable escalation in cellular proliferation, positive staining for alpha-smooth muscle actin (α-SMA), and an augmented production of extracellular matrix (ECM) proteins. Thus, the activation of HSCs plays a vital role in the development and progression of liver fibrosis [7]. Cells resembling HSCs were isolated also from the pancreas, and similar to HSCs, these pancreatic stellate cells (PSCs) play an important role in pancreatic fibrogenesis [8]. Previous studies have suggested that cells containing vitamin A-storing lipid droplets are also present in the lungs and kidneys [9,10], and we recently isolated and characterized stellate cells from the kidneys [11]. Therefore, stellate cells are considered an attractive target for anti-fibrotic therapies [12]. However, the exploration of potential stellate cells in lung tissue has remained unexplored thus far.

In our previous studies, we found that albumin expression was detectable in quiescent stellate cells but absent in activated cells, although its expression level in stellate cells was relatively low compared to that in liver cells [13]. Albumin, primarily synthesized in hepatocytes, is the most abundant protein in plasma and serves multiple functions, including regulating plasma oncotic pressure and facilitating the transportation of various substances [14]. This protein consists of three structurally similar domains (I, II, and III) and exhibits binding affinity for various hydrophobic ligands, including fatty acids and retinoids [15,16]. Interestingly, when albumin was forcibly expressed in activated stellate cells, it triggered a remarkable phenotypic reversal of myofibroblasts to an early activated cell state [13]. This resulted in the reoccurrence of cytoplasmic lipid droplets and a noticeable reduction in the expression of α-SMA and collagen type I.

Based on these findings, a novel recombinant fusion protein (referred to as R-III) was developed as an effective anti-fibrotic agent [17]. R-III consists of the albumin domain III fused to the C-terminus of retinol-binding protein (RBP). The selection of RBP for targeted delivery to stellate cells stemmed from its pivotal role in facilitating the cellular uptake of retinol within HSCs via interaction with the membrane receptor called STRA6 [18]. Subsequent investigations demonstrated that R-III exhibited inhibitory effects on the activation of hepatic, pancreatic and renal stellate cells in vitro [11,17,19]. Moreover, in vivo studies revealed that R-III contributed to a reduction in liver and kidney fibrosis [11,20]. Therefore, the present study aimed to determine the therapeutic effects of R-III on bleomycin-induced pulmonary fibrosis and examine the presence of stellate cells in lung tissues.

## 2. Materials and Methods

### 2.1. Materials

Male Sprague-Dawley rats, aged eight weeks, and male BALB/c mice, aged seven weeks, were obtained from OrientBio, Inc. (Seongnam, Republic of Korea) and kept in a controlled environment with regulated temperature, humidity, and lighting conditions. The animal experiments received approval from our institutional review board and adhered to the guidelines outlined in the Guide for the Care and Use of Laboratory Animals. The synthesis of the fusion protein R-III (depicted in Appendix A) followed a method previously described [17]. Briefly, CHO cells were transfected with an R-III expression vector to establish stable expression. The purification of the secreted His-tagged R-III involved Ni-NTA affinity resin and size-exclusion chromatography.

### 2.2. Mouse Model of Bleomycin-Induced Pulmonary Fibrosis

The bleomycin mouse model was chosen due to two reasons: firstly, it significantly reduced the required amount of R-III for crucial in vivo experiments, and secondly, a meta-analysis study of the bleomycin model indicates that both mouse and rat models contribute equally to the study of lung fibrosis [21]. Bleomycin sulfate (Santa Cruz, Santa Cruz, CA, USA) was dissolved in sterile PBS solution. Seven-week-old male BALB/c mice were anesthetized with inhalational isoflurane using an isoflurane vaporizer, and a single 60 µL aliquot containing 0.08 U of bleomycin diluted in normal saline was injected intratracheally. To determine the therapeutic effect of R-III, mice (*n* = 14) treated with bleomycin were randomly divided into two groups and injected via the tail vein with either saline or R-III (15 µg) every other day, starting on day two after the bleomycin injection. Mice were sacrificed 12 days after bleomycin administration, and their lungs were collected for analysis. Mice (*n* = 7) instilled with saline were used as controls. All experiments were performed in duplicate.

### 2.3. Immunohistochemical Analysis

After routine processing, 5-µm-thick sections of formalin-fixed, paraffin-embedded lung tissues were prepared and stained with hematoxylin plus eosin for histological analysis. Tissue sections were also immunohistochemically stained with the following antibodies: α-SMA (Abcam #ab32575, Cambridge, MA, USA), collagen type I (Abcam #ab292), His-tag (Abcam #ab84162), and TGF-β (Santa Cruz #sc146, Santa Cruz, CA, USA).

### 2.4. Hydroxyproline Measurement

Lung hydroxyproline levels were measured according to the manufacturer’s protocol (Sigma-Aldrich, St. Louis, MO, USA). Briefly, lung tissue was homogenized in distilled water and mixed with an equal volume of concentrated hydrochloric acid (~12 N HCl), and the homogenates were incubated at 120 °C for 3 h. After hydrolysis, 5 μL of each sample was oxidized with chloramine T (Sigma-Aldrich), followed by an enzymatic reaction with 4-dimethylaminobenzaldehyde (DMAB) solution. The sample absorbance was measured in duplicate at 560 nm, and the hydroxyproline concentration was calculated from a standard curve using known concentrations of hydroxyproline.

### 2.5. Isolation of Rat Lung Stellate Cells (LSCs) and Cell Culture

LSCs were obtained from male Sprague–Dawley rats, aged eight weeks, following the previously established protocol [13]. The process involved in situ perfusion of the lungs with phosphate-buffered saline (PBS), followed by Hank’s buffered salt solution (HBSS) enriched with collagenase, pronase (Sigma-Aldrich), and DNase (MP Biomedicals, Santa Ana, CA, USA) through the right ventricles. Subsequently, the perfused lungs were dissected and subjected to further digestion in a 37 °C water bath using HBSS supplemented with collagenase, pronase, and DNase for a duration of 10 min. The resulting cell suspension was then washed and centrifuged on a 13.4% Nycodenz gradient at 1400× *g* for 20 min without braking. The collected interface, containing the LSCs, underwent additional washing with HBSS. Finally, the isolated LSCs were cultured in DMEM supplemented with 10% FBS and examined through microscopic observation.

### 2.6. Immunofluorescence

The LSCs were carefully placed onto gelatin-coated glass coverslips. Subsequently, the cells were fixed using a 4% paraformaldehyde solution and permeabilized with PBS containing 0.1% Triton X-100. Next, the samples were subjected to an overnight incubation at 4 °C with the primary antibodies, suitably diluted in 1% BSA in PBST (PBS with 0.1% Tween 20). This was followed by exposure to the FITC- or Alexa Fluor 594-conjugated secondary antibodies. In this experiment, the primary antibodies used were anti-albumin (Affinity Bioreagents, Rockford, IL, USA) and anti-α-SMA (Sigma-Aldrich). The LSCs were observed using a Leica TCS SP8 microscope.

### 2.7. Quantitative Real-Time PCR

Total RNA isolation was carried out using TRIzol (Ambion, Austin, TX, USA), and the isolated RNA was then used for cDNA synthesis. Real-time PCR was conducted using the ABI QuantStudio^TM^ 3 Real-Time PCR system. To account for reaction variability, the PCR products were normalized by comparing them to the mRNA levels of glyceraldehyde 3-phosphate dehydrogenase (*GAPDH*). The primers used were 5′-TATCTGGGAAGGGCAGCAAA-3′ (forward primer) and 5′-CCAGGGAAGAAGAGGAAGCA-3′ (reverse primer) for α-SMA; 5′-GGAGAGTACTGGATCGAC-3′ (forward) and 5′-CTGACCTGTC TCCATGTT-3′ (reverse) for collagen type I; and 5′-GGTGGTCTCCTCTGACTTCAACA-3′ (forward) and 5′-GTTGCTGTAGCCAAATTCGTTGT-3′ (reverse) for GAPDH.

### 2.8. Statistical Analysis

The obtained results were presented as the means ± standard deviation (SD). In vitro data analysis was conducted using a paired t-test. On the other hand, the Kruskal–Wallis test followed by the Dwass–Steel–Critchlow–Fligner (DSCF) multiple comparison test was employed for analyzing the data from in vivo studies. Statistical analyses were carried out using SPSS for Windows, version 20.0 (SPSS, Inc., Chicago, IL, USA). A significance level of *p* < 0.05 was considered statistically significant.

## 3. Results

### 3.1. R-III Administration Reduced Bleomycin-Induced Lung Fibrosis

To investigate the therapeutic effects of R-III on bleomycin-induced lung fibrosis, BALB/c mice were intratracheally instilled with bleomycin. They were then randomly divided into two groups and administered either saline or R-III (15 µg) via the tail vein every other day, starting on day two after the bleomycin injection. The body weight of the mice was measured daily. Mice treated with bleomycin displayed a decrease in body weight, and weight regain began after the ninth day after instillation. Notably, mice treated with both bleomycin and R-III began to regain weight earlier than those treated with bleomycin alone (Appendix A). Lung sections from control mice instilled with saline showed no histopathological changes, whereas lung sections from the bleomycin-treated mice exhibited significant histopathological alterations, including the presence of extensive fibrous regions, collapsed alveolar spaces, and structural disruption of lung tissues (Figure 1A, upper panel). Immunohistochemistry confirmed excessive collagen type I deposition as well as intense immunostaining for α-SMA and the pro-fibrogenic mediator TGF-β in the bleomycin-treated mice (Figure 1B). However, the lungs of mice treated with both bleomycin and R-III showed amelioration of lung fibrosis and comparatively preserved lung architecture (Figure 1A, upper right panel). Immunoreactivity with collagen, α-SMA, and TGF-β was also significantly reduced with R-III treatment. The impact of R-III on lung fibrosis was further evaluated using a hydroxyproline assay (Figure 2). The hydroxyproline content in the lungs of the bleomycin-treated mice showed an approximately twofold increase compared to that of the control mice, which was then reduced by approximately 35% with R-III treatment.

### 3.2. Intravenously Injected R-III Was Delivered to Autofluorescent Cells in Lung Tissue

We then performed immunohistochemistry using an antibody against the His-tag to examine the cellular distribution of injected His-tagged R-III in the lung. The findings revealed strong staining in fibrotic foci, although a weak, non-specific signal was also observed (Appendix A). In order to gain a better understanding, we administered a single intravenous dose of saline or R-III to BAL/c mice. Three days after injection, the lungs were harvested and preserved by freezing them at optimum cutting temperature (OCT). Subsequently, when we illuminated the lung sections with a UV wavelength of 330–360 nm, we observed rapid-fading, vitamin A-specific autofluorescence (Figure 3). This autofluorescence exhibited an aggregated, speckled pattern surrounding the nuclei. This finding supports a previous report indicating the presence of vitamin A-storing cells in the lung [9]. Additionally, immunofluorescence using an anti-His-tag antibody demonstrated a pronounced localization of the His-tag signal within the autofluorescent cells.

### 3.3. Cells Resembling HSCs Are Present in Lung Tissue

A previous study suggested the presence of vitamin A-storing cells in the lungs [9]. In our study, we have shown the effectiveness of R-III, a drug specifically designed to target stellate cells, in reducing lung fibrosis. These findings prompted us to isolate the pulmonary counterparts of HSCs. By making a minor modification to the HSC isolation protocol, we attempted to isolate these cells from the mouse lung but were unsuccessful. However, we were able to isolate approximately 5 × 10^3^ cells per rat lung (referred to as lung stellate cells, LSCs) and cultured them in dishes. LSCs demonstrated a slow attachment process, resembling the sluggish attachment observed in stellate cells derived from the liver, pancreas, and kidney. By day 3 after seeding (referred to as LSCs d3), LSCs exhibited stable adhesion to the culture plate. Additionally, they displayed a flattened polygonal shape and contained cytoplasmic lipid droplets, similar to the observations made in hepatic, pancreatic, and renal stellate cells on day 3 after seeding (Figure 4A) [7,8,11]. Like stellate cells found in other tissues, LSCs spontaneously underwent activation/transdifferentiation when cultured on plastic in vitro. Following the second passage (LSCs P2), LSCs fully transformed into myofibroblast-like cells, leading to the depletion of their lipid droplets (Figure 4A). Confocal microscopy revealed that LSCs d3, but not LSCs P2, exhibited an albumin signal and displayed vitamin A-specific autofluorescence at 330–360 nm, both of which exhibited an aggregated, speckled pattern (Figure 4B). In contrast, α-SMA expression was detected in LSCs P2. Interestingly, when LSCs P2 were treated with R-III, they underwent a remarkable transformation into distinct cell types, characterized by the reappearance of autofluorescent lipid droplets and reduced α-SMA expression. This transformation was reminiscent of the R-III-induced inactivation of hepatic and renal stellate cells [11,20]. Real-time PCR analysis also revealed that the mRNA expression levels of collagen type I and α-SMA, both well-known markers for activated HSCs/myofibroblasts, were elevated in LSCs P2 compared to those in LSCs d3 (Figure 5A). Conversely, treatment of LSCs P2 with R-III resulted in decreased mRNA expression levels of these markers (Figure 5B). These findings suggest that the cells isolated from lung tissue are pulmonary counterparts of HSCs and that the in vivo anti-fibrotic effect of R-III is probably due to its action on LSCs.

## 4. Discussion

Tissue fibrosis is characterized by the excessive accumulation of extracellular matrix (ECM), resulting in the disruption of organ function and posing a risk to the patient’s life. It can occur in different locations throughout the human body, with liver fibrosis being the most extensively studied among fibrotic tissues, followed by pancreatic, renal, and pulmonary fibrosis. The traditional perspective has long held that tissue fibroblasts and epithelial–mesenchymal transition (EMT) are the primary sources of myofibroblasts [22]. Although bone marrow-derived stem cells and endothelial–mesenchymal transition (EndoMT) have been proposed to have a minor role in this process, their significance is still under debate. Stellate cells, isolated from the liver and pancreas in the late 1980s and late 1990s, respectively, were extensively studied and found to play an important role in the fibrosis of their respective sites [7,8]. Hepatic, pancreatic, and renal stellate cells reside in specific locations such as the sinusoidal space, periacinar/perivascular/ periductal regions, and perivascular/peritubular region, respectively. These vitamin A-storing cells exhibit striking similarities not only in morphology and perivascular location but also in the molecular fibrotic pathway [7,8,11]. The activation of stellate cells into myofibroblasts, which plays a crucial role in tissue fibrogenesis, can be readily evaluated by the upregulation of α-SMA and collagen type I expression.

Recent studies utilizing genetic lineage mapping have provided insights into the involvement of pericytes in the fibrogenesis of the kidneys and lungs [3,23]. Another line of research has also suggested that pulmonary lipofibroblasts, located in the alveolar interstitium, could potentially serve as a source of myofibroblasts in lung fibrosis [24,25]. These lipofibroblasts demonstrate the ability to undergo phenotypic plasticity, transitioning between lipogenic and myogenic states during the formation and resolution of fibrosis. This phenomenon of phenotypic switching has been extensively studied in hepatic and pancreatic stellate cells, where cytoplasmic lipid droplets are involved in vitamin A storage [8,26]. Building upon these reports and our own in vivo findings, we proceeded to isolate lung stellate cells (LSCs) from rat lung tissue. Our investigation revealed that rat LSCs displayed comparable morphological and biochemical characteristics to those observed in hepatic, pancreatic, and renal stellate cells [7,8,11]. LSCs at day 3 (LSCs d3) displayed cytoplasmic lipid droplets that emitted autofluorescence and showed minimal or negligible expression of α-SMA and collagen type I. These characteristics resembled those of early-activated stellate cells. In contrast, LSCs at passage 2 (LSCs P2) exhibited a myofibroblastic phenotype with elevated levels of α-SMA and collagen expression. This transition towards myofibroblast-like cells during in vitro cell culture reflects the activation process observed in stellate cells found in other tissues. These findings suggest that the cells we isolated are the pulmonary stellate cells. Further study is needed to determine the optimal isolation protocol for LSCs and to thoroughly characterize their properties.

Our attempts to isolate LSCs from mice were unsuccessful in achieving sufficient purity. This could be attributed to technical challenges or the possibility of a limited cell population, or possibly both. It is worth noting that disparities in stellate cells exist among different species. For instance, the isolation of mouse HSCs is considerably more challenging compared to rat HSCs. Mouse HSCs have a smaller cell size compared to their rat counterparts. Furthermore, the maximum number of cell passages for mouse HSCs is generally lower compared to that observed in rat cells.

The existence of vitamin A-storing stellate cells in various tissues is still not fully understood. When tissues are injured, a prompt initiation of the wound healing response is essential. Conversely, myofibroblasts, which are key participants in the wound healing process, need to disappear as the wound heals. Stellate cells, when cultured on plastic or exposed to TGF-β, undergo a transformation into activated/myofibroblastic cells. However, they readily revert to an early-activated cell phenotype in response to different stimuli, such as albumin, PPAR-γ, and C/EBP expression [13,27]. Consequently, stellate cells can contribute to the timely progression of wound healing, underscoring their importance in diverse tissues. Nevertheless, there is speculation that the prolonged activation of stellate cells in response to chronic tissue injury may disrupt their regulation, leading to an excessive deposition of extracellular matrix (ECM) in the affected tissue.

Stellate cells have emerged as a promising target for anti-fibrotic therapies, and numerous previous studies have focused on strategies such as inducing inactivation, apoptosis, and senescence of activated stellate cells to mitigate tissue fibrosis. However, currently, there is a lack of effective therapies available for the treatment of tissue fibrosis. This could be attributed to our limited understanding of the mechanisms underlying stellate cell activation and the absence of target-specific candidate drugs [28]. Despite extensive research conducted over the past three decades, the molecular mechanism responsible for stellate cell activation remain elusive. For instance, quiescent stellate cells store retinoids as retinyl esters within cytoplasmic lipid droplets. As the activation process progresses, there is a rapid depletion of lipid droplets, potentially resulting in the release and metabolism of a portion of the retinoid content, leading to the production of retinoic acid (RA). The involvement of retinoids in HSC activation has been proposed; however, previous studies examining the effects of exogenous retinoids on HSC activation and liver fibrosis have produced conflicting results [29].

In this study, we have demonstrated that R-III induces phenotypic switching of activated/myofibroblastic LSCs into a lipogenic, early-activated phenotype, which may contribute to its in vivo anti-fibrotic effect.

## 5. Conclusions

Our results demonstrate that R-III, a stellate cell-targeted anti-fibrotic drug, effectively mitigates bleomycin-induced lung fibrosis. Notably, we successfully isolated pulmonary stellate cells, which exhibited similar morphological and biochemical characteristics to stellate cells found in the liver, pancreas, and kidneys. Based on these results, R-III holds promise as a potential novel candidate for an anti-fibrotic drug.

## Figures and Tables

**Figure 1 biomedicines-11-02007-f001:**
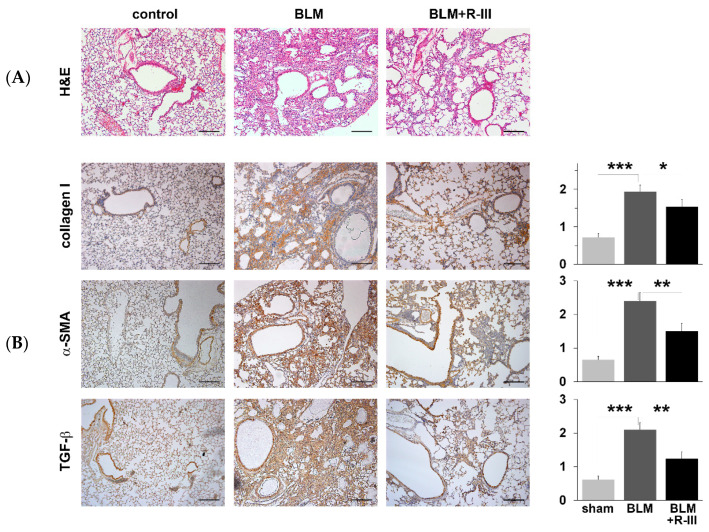
Fusion protein (R-III) of retinol-binding protein and albumin domain III reduces bleomycin (BLM)-induced lung fibrosis. Lung sections from mice treated with control, BLM, and BLM + R-III (each group = 7 mice) were stained with hematoxylin plus eosin (**A**) and subjected to immunohistochemistry against collagen type I, α-SMA, and TGF-β (**B**). Representative images from each study group are shown. Semiquantitative analysis of the staining intensity for each group is shown (right). Data are expressed as the means ± SD (*p*-value; Kruskal–Wallis test, followed by DSCF multiple comparison test). * *p* < 0.05, ** *p* < 0.01, *** *p* < 0.001. Scale bar, 200 μm.

**Figure 2 biomedicines-11-02007-f002:**
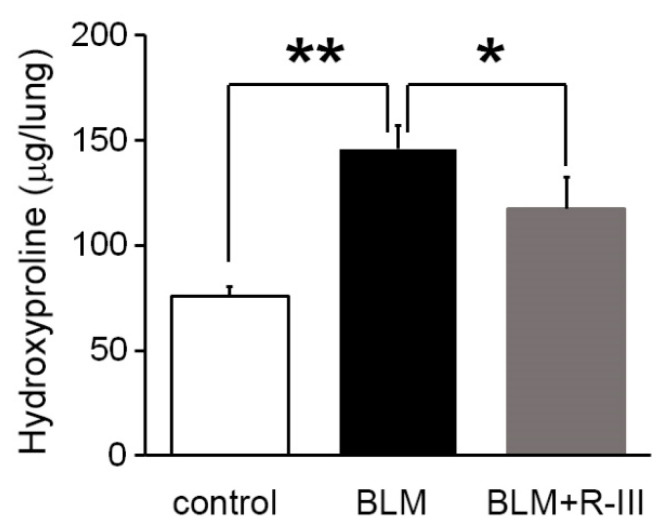
R-III reduces collagen deposition in the lungs induced by BLM. The lungs of mice treated with control, BLM, and BLM + R-III (each group = 7 mice) were homogenized and hydrolyzed. Subsequently, the collagen content was measured using the hydroxyproline assay. The data are presented as means ± SD, and statistical significance was determined using the Kruskal–Wallis test followed by DSCF multiple comparison test (*p*-value). The results indicate that * *p* < 0.05 and ** *p* < 0.01.

**Figure 3 biomedicines-11-02007-f003:**
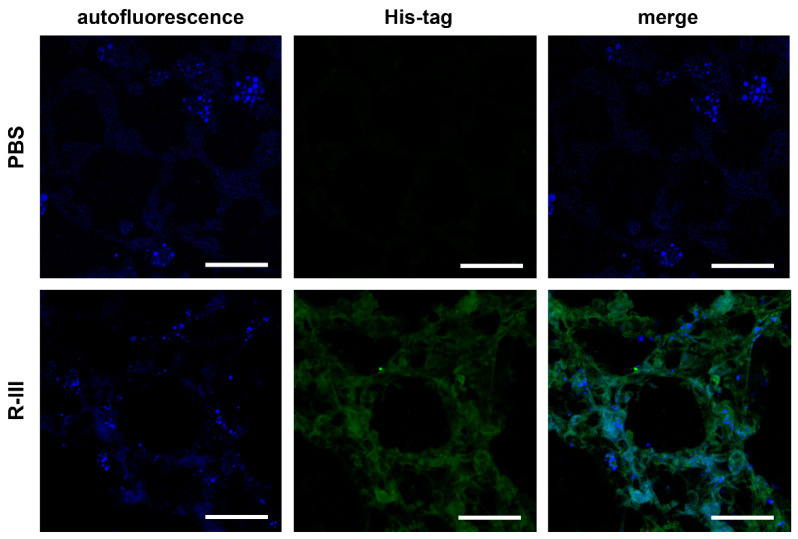
R-III is delivered to autofluorescent cells in lung tissue. Mice were euthanized three days after intravenous injection of saline or R-III (each group = 3 mice), and 18 μm lung sections were obtained by slicing OCT-embedded lungs and were subsequently analyzed by immunofluorescence. The images displayed include autofluorescent images, immunofluorescence images using an anti-His-tag antibody, and merge images. Scale bar, 50 μm.

**Figure 4 biomedicines-11-02007-f004:**
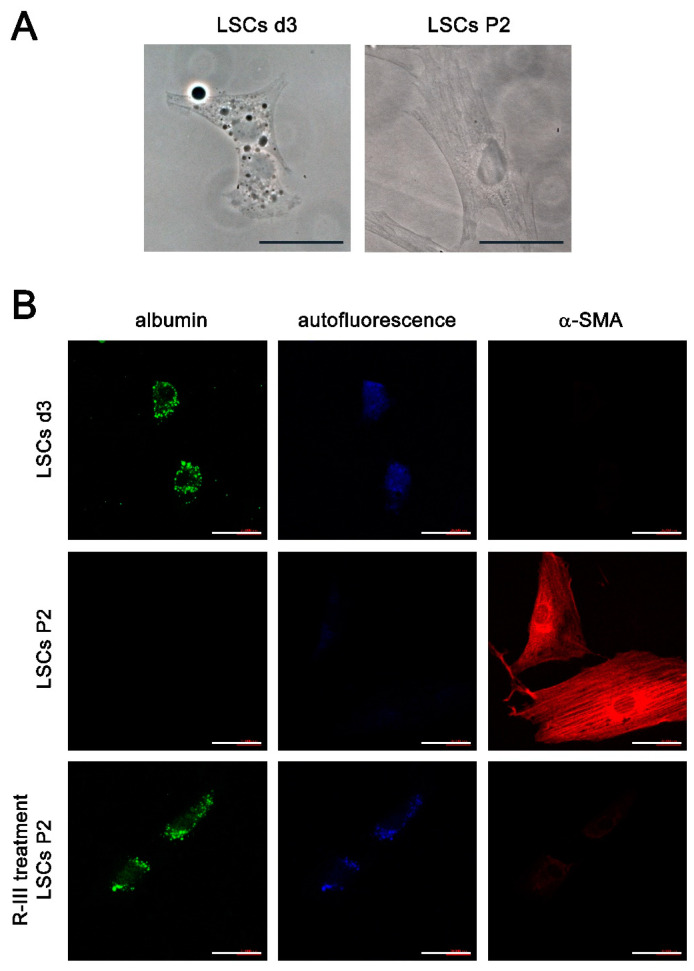
Lung stellate cells (LSCs) are present in lung tissues. (**A**) Phase-contrast images are presented for LSCs on day 3 after seeding (LSCs d3) and LSCs after passage 2 (LSCs P2). (**B**) The displayed images include immunofluorescence images using antibodies against albumin or α-SMA, as well as autofluorescence images for LSCs d3, LSCs P2, and LSCs P2 treated with R-III, respectively. Scale bar, 40 μm.

**Figure 5 biomedicines-11-02007-f005:**
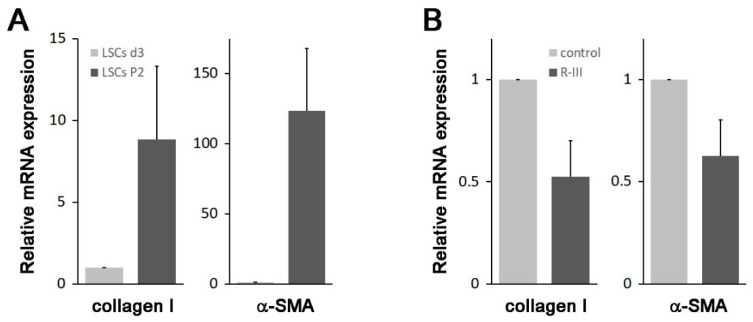
Changes in the expression levels of collagen type I and α-SMA. (**A**) Total RNA was extracted from LSCs on day 3 after seeding (*n* = 3 rats) and after passage 2 (*n* = 1 rat). Subsequently, the expression levels of collagen type I and α-SMA were analyzed using real-time PCR. (**B**) LSCs P2, cultured with 180 μL of media, were treated with PBS (20 μL) or R-III (20 μL of 0.3 mg/mL R-III; final conc. 0.75 μM) for 20 h and analyzed by real-time PCR. The data represent the means ± SD for two independent experiments.

## Data Availability

The data presented in this study are available on request from the corresponding author.

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
