# Peer review of "Fusion Protein of RBP and Albumin Domain III Reduces Lung Fibrosis by Inactivating Lung Stellate Cells"

_biomedicines, 2023, doi:10.3390/biomedicines11072007_

Round 1
Reviewer 1 Report
This is an interesting studying applying R-III administration for against bleomycin-induced lung fibrogensis in mice. Reduced level of collaneg, aSMA, and TGFb were identified after the treatment of R-III.
The description of experiments, especially the concept between HSC and LSC should be more clear in the subsection of 3.3. The authors have identified HSC plays an important role in liver fibrosis, but this article is focusing on lung. No direct relevant to liver fibrosis here, or the authors have used HSC in this model. Please clarify. The same concept applies to the introduction, instead of introducing HSC and PSC, should more focus on the role of LSC during fibrogenesis.
Please also explain clearly of the rationale comparing the LSC at day 3 culture to passage 2. Is there any scientific meaning of this comparison?
Other minor suggestions for the author to consider as well:
1. in figure 4 and 5B, the autofluorescence should blue, is that means DAPI staining for nuclei? Please identify rather that using the word of autofluorescence.
2. In figure 3, if this figure is aiming to showing the logic of using frozen section to detect the distribution His-Tag, it will be better to leave in support info rather than in main text.
3. Please provide sample volume in figure legend.
4. Please clear the results sections with limited ref citation, and focus a bit more to discussion section.
5. Please consider higher magnification of IHC or H&E pictures rather the 4x .
English writing looks fine but the way of wording throughout the manuscript needs improvement.
Author Response
Response to Reviewer 1 Comments
Point 1: The description of experiments, especially the concept between HSC and LSC should be more clear in the subsection of 3.3. The authors have identified HSC plays an important role in liver fibrosis, but this article is focusing on lung. No direct relevant to liver fibrosis here, or the authors have used HSC in this model. Please clarify. The same concept applies to the introduction, instead of introducing HSC and PSC, should more focus on the role of LSC during fibrogenesis.
Response 1: The reviewer's comment is valid, emphasizing the need for our manuscript to focus on lung fibrosis and lung stellate cells. We have extensive experience working with hepatic, pancreatic, and renal stellate cells over the years. However, in this study, we have successfully isolated stellate cells from the lungs for the first time and conducted a thorough characterization by comparing them to stellate cells found in other tissues. In response to this feedback, we have rewritten the introduction and section 3.3 to highlight the properties of LSCs and their similarities with stellate cells from various tissues.
Point 2: Please also explain clearly of the rationale comparing the LSC at day 3 culture to passage 2. Is there any scientific meaning of this comparison?
Response 2: Similar to hepatic and pancreatic stellate cells, LSCs displayed a slow attachment process. By day 3 after seeding (referred to as LSCs d3), LSCs demonstrated stable adhesion to the culture plate. Moreover, LSCs underwent spontaneous activation/transdifferentiation when cultured on plastic in vitro. Upon reaching the second passage (LSCs P2), LSCs fully transformed into myofibroblast-like cells. This is why stellate cells on day 3 after seeding and after passage 2 are typically compared. We have included this information in the results of the revised manuscript (page 6, line 246).
Point 3: in figure 4 and 5B, the autofluorescence should blue, is that means DAPI staining for nuclei? Please identify rather than using the word of autofluorescence.
Response 3: Vitamin A, which possesses conjugated systems with alternating double and single bonds, exhibits fluorescence at a UV wavelength of 330-360 nm. In Figure 3 and 4B, LSCs were not stained with DAPI, and the blue-colored signals displaying a speckled pattern represent vitamin A-specific autofluorescence. It is important to note that the mention of DAPI staining in the materials and methods was an error, which has been removed in the revised manuscript. Additionally, we have magnified the images in Figure 3, enabling easy visualization of the autofluorescence images. Lastly, in the revised manuscript, we have included the term "vitamin A-specific autofluorescence" for clarification (page 6 line 224; page 7 256).
Point 4: In figure 3, if this figure is aiming to showing the logic of using frozen section to detect the distribution His-Tag, it will be better to leave in support info rather than in main text.
Response 4: As per your request, we have relocated Figure 3 from the main text to the supplementary information section.
Point 5: Please provide sample volume in figure legend.
Response 5: According to your request, we have now included the sample volume for Figure 2 and 5 in the revised manuscript.
Point 6: Please clear the results sections with limited ref citation, and focus a bit more to discussion section.
Response 6: As per your request, we have moved the initial portion of section 3.3 to the Discussion section.
Point 7: Please consider higher magnification of IHC or H&E pictures rather the 4x .
Response 7: As per your request, we have replaced the images with X10 magnified images.
Reviewer 2 Report
1. The title partially reflects the content of the article.
2. In the "Abstract" section, the authors briefly presented the relevance and purpose of the article. Stellate cells are involved in fibrosis of the liver, pancreas and kidneys. The authors suggested that stellate cells are localized in the lungs. In addition, the authors suggested the participation of stellate cells in the effects of the fused R-III protein in laboratory animals on a model of bleomycin-induced lung fibrosis. The studies were conducted in vitro and in vivo. The section briefly presents the results of the study, on the basis of which the authors concluded that R-III can reduce lung fibrosis by inactivating stellate cells.
3. The "Keywords" presented in the article are necessary. Recommendation: it is more correct to specify as the object of the study not albumins, but a specific protein R-III.
4. In the section "1. Introduction" the authors briefly characterized idiopathic pulmonary fibrosis (IPF), various cell populations that act as a source of collagen. The authors are particularly interested in pericytes (stellate cells), since stellate cells are considered an attractive target for antifibrotic therapy. In this paper, the authors set a goal to isolate selected cells from the lungs of laboratory animals and to show the therapeutic effect of R-III on bleomycin-induced pulmonary fibrosis. Meanwhile, the title of the article display only the pharmacological part of the study. The search for stellate cells in the lung tissue of laboratory animals of two species is not displayed. I ask the authors to pay attention to this circumstance and make adjustments to the title of the article.
5. In the section "2. Materials and methods of research" presented laboratory animals conditions and their maintenance, the approval of the study by the institutional supervisory Board and conformed to the Guidelines for the Care of Laboratory Animals and their Use. In the section "2. Materials and methods" describes a model of lung fibrosis, immunohistochemical analysis, measurement of hydroxyproline content, isolation of rat lung stellate cells (LSCS) and cell culture, immunofluorescence, quantitative real-time PCR and statistical analysis.
6. In the "Results" section, the authors consistently demonstrate the results of their study: therapeutic effects of R-III in bleomycin-induced pulmonary fibrosis were revealed in mice, and evidence of R-III delivery to autofluorescent cells in lung tissue is also presented. In addition, the authors revealed the presence of cells resembling HSCs in the lung tissue of rats. The results of the study are presented in sufficient volume for their evaluation, well described, accompanied by drawings.
7. It follows from the text of the article that mice were treated with bleomycin, rats were not treated with bleomycin. If this is my understanding of reality, then questions arise. Thus, from the results presented in the "Results" section, it follows that rats are a convenient model for studying stellate cells and R-III effects with the participation of stellate cells: in rats, cells resembling HSCs were identified, the effects of R-III were studied on the culture of rat stellate cells and there is an assumption that stellate cells act as a target for R- III. These data imply the modeling of lung fibrosis in rats and the study of the therapeutic effects of R-III in rats with lung fibrosis. However, this did not happen. The authors presented studies of the effects of R-III in mice with lung fibrosis and the negative experience of stellate cell isolation in mice. Please explain what this situation is related to?
8. In the Discussion section, the authors discussed the results obtained with the involvement of literature and analyzed the possibility of using R-III to target stellate cells. R-III may serve as a new candidate for an antifibrotic drug. An interesting conclusion is that stellate cells are present in the lung and these cells may be a potential target for III.
9. There is no "CONCLUSION" section in the article. I recommend adding the "CONCLUSION" section to the article.
10. The text of the article is clearly written. The perception of this article is improved by drawings. All drawings are clear.
11. The article is interesting and important. The manuscript did not cause any ethical problems. However, the perception of the text would improve if the authors presented the design of the study in the section "Materials and Methods". I recommend doing this. The authors pointed out the difficulties associated with the detection of stellate cells in the lungs of mice. Meanwhile, the experience of isolating stellate cells from rat lungs was positive. In this regard, it was logical to conduct studies not only on rat lung cell culture in vitro, but also to evaluate the therapeutic effects of R-III on an in vivo fibrosis model. This was not done, which reduces the significance of this study.
12. All links to publications in the "References" section are necessary and correct, made in the right style. Of the 25 links that are presented in the article, only 2 links have been in the last 5 years. It is necessary to add additional links on the topic of the article for the last 5 years. Authors should pay attention to the fact that the numbering of links is shown twice. In this regard, it is necessary to make appropriate adjustments to the "References" section.
13. I have no concerns about the similarity of this article with other articles published by the same authors.
14. Competing interests of authors do not create bias in the presentation of results and conclusions.
Author Response
Response to Reviewer 2 Comments
Point 1: The title partially reflects the content of the article.
Response 1: As per your request, we have modified the title to accurately reflect the content of the manuscript.
Point 2: In the "Abstract" section, the authors briefly presented the relevance and purpose of the article. Stellate cells are involved in fibrosis of the liver, pancreas and kidneys. The authors suggested that stellate cells are localized in the lungs. In addition, the authors suggested the participation of stellate cells in the effects of the fused R-III protein in laboratory animals on a model of bleomycin-induced lung fibrosis. The studies were conducted in vitro and in vivo. The section briefly presents the results of the study, on the basis of which the authors concluded that R-III can reduce lung fibrosis by inactivating stellate cells.
Response 2: We appreciate the valuable comment provided by the reviewer.
Point 3: The "Keywords" presented in the article are necessary. Recommendation: it is more correct to specify as the object of the study not albumins, but a specific protein R-III.
Response 3: As per your request, we have modified the keywords accordingly.
Point 4: In the section "1. Introduction" the authors briefly characterized idiopathic pulmonary fibrosis (IPF), various cell populations that act as a source of collagen. The authors are particularly interested in pericytes (stellate cells), since stellate cells are considered an attractive target for antifibrotic therapy. In this paper, the authors set a goal to isolate selected cells from the lungs of laboratory animals and to show the therapeutic effect of R-III on bleomycin-induced pulmonary fibrosis. Meanwhile, the title of the article display only the pharmacological part of the study. The search for stellate cells in the lung tissue of laboratory animals of two species is not displayed. I ask the authors to pay attention to this circumstance and make adjustments to the title of the article.
Response 4: As per your request, we have modified the title to accurately reflect the content of the manuscript.
Point 5: In the section "2. Materials and methods of research" presented laboratory animals conditions and their maintenance, the approval of the study by the institutional supervisory Board and conformed to the Guidelines for the Care of Laboratory Animals and their Use. In the section "2. Materials and methods" describes a model of lung fibrosis, immunohistochemical analysis, measurement of hydroxyproline content, isolation of rat lung stellate cells (LSCS) and cell culture, immunofluorescence, quantitative real-time PCR and statistical analysis.
Response 5: We appreciate the valuable comment provided by the reviewer.
Point 6: In the "Results" section, the authors consistently demonstrate the results of their study: therapeutic effects of R-III in bleomycin-induced pulmonary fibrosis were revealed in mice, and evidence of R-III delivery to autofluorescent cells in lung tissue is also presented. In addition, the authors revealed the presence of cells resembling HSCs in the lung tissue of rats. The results of the study are presented in sufficient volume for their evaluation, well described, accompanied by drawings.
Response 6: We appreciate the valuable comment provided by the reviewer.
Point 7: It follows from the text of the article that mice were treated with bleomycin, rats were not treated with bleomycin. If this is my understanding of reality, then questions arise. Thus, from the results presented in the "Results" section, it follows that rats are a convenient model for studying stellate cells and R-III effects with the participation of stellate cells: in rats, cells resembling HSCs were identified, the effects of R-III were studied on the culture of rat stellate cells and there is an assumption that stellate cells act as a target for R- III. These data imply the modeling of lung fibrosis in rats and the study of the therapeutic effects of R-III in rats with lung fibrosis. However, this did not happen. The authors presented studies of the effects of R-III in mice with lung fibrosis and the negative experience of stellate cell isolation in mice. Please explain what this situation is related to?
Response 7: Our current supply of highly purified, recombinant R-III reagent is limited. A Meta-analysis study of the bleomycin model in both rat and mouse models (Moeller et al., The International Journal of Biochemistry and Cell Biology 2008, 40:361. doi:10.1016/j.biocel.2007.08.011.) suggests that these models contribute equally to the study of lung fibrosis. We used the bleomycin mouse model because this significantly reduced the amount of albumin-RBP fusion protein (R-III) needed for these critical in vivo experiments. We included this in the materials and methods of the revised manuscript (page 2, line 96).
Point 8: In the Discussion section, the authors discussed the results obtained with the involvement of literature and analyzed the possibility of using R-III to target stellate cells. R-III may serve as a new candidate for an antifibrotic drug. An interesting conclusion is that stellate cells are present in the lung and these cells may be a potential target for III.
Response 8: We appreciate the valuable comment provided by the reviewer.
Point 9: There is no "CONCLUSION" section in the article. I recommend adding the "CONCLUSION" section to the article.
Response 9: As per your request, we have now included the conclusion in the revised manuscript.
Point 10: The text of the article is clearly written. The perception of this article is improved by drawings. All drawings are clear.
Response 10: We appreciate the valuable comment provided by the reviewer.
Point 11: The article is interesting and important. The manuscript did not cause any ethical problems. However, the perception of the text would improve if the authors presented the design of the study in the section "Materials and Methods". I recommend doing this. The authors pointed out the difficulties associated with the detection of stellate cells in the lungs of mice. Meanwhile, the experience of isolating stellate cells from rat lungs was positive. In this regard, it was logical to conduct studies not only on rat lung cell culture in vitro, but also to evaluate the therapeutic effects of R-III on an in vivo fibrosis model. This was not done, which reduces the significance of this study.
Response 11: As per your request, we have now included the rationale for using the bleomycin mouse model in the materials and methods section of the revised manuscript (page 2, line 96).
Point 12: All links to publications in the "References" section are necessary and correct, made in the right style. Of the 25 links that are presented in the article, only 2 links have been in the last 5 years. It is necessary to add additional links on the topic of the article for the last 5 years. Authors should pay attention to the fact that the numbering of links is shown twice. In this regard, it is necessary to make appropriate adjustments to the "References" section.
Response 12: As per your request, we have included recent references in the revised manuscript. We have cited six references published within the past five years, along with four additional references published in 2017. Additionally, we have corrected the reference style according to the specified guidelines.
Point 13: I have no concerns about the similarity of this article with other articles published by the same authors.
Response 13: We appreciate the valuable comment provided by the reviewer.
Point 14: Competing interests of authors do not create bias in the presentation of results and conclusions.
Response 14: We appreciate the valuable comment provided by the reviewer.
Round 2
Reviewer 1 Report
The authors have answered most of my concerns.
However, the sample volume is still missing for individual figures or this study, the sample volume means how many mice/samples have been used in individual groups/experiments.
Author Response
Response to Reviewer 1 Comments
Point 1: The authors have answered most of my concerns. However, the sample volume is still missing for individual figures or this study, the sample volume means how many mice/samples have been used in individual groups/experiments.
Response 1: The reviewer's comment is valid, as we inadvertently omitted the number of animals used in the manuscript. In response to your request, we have now included the sample volume for Figures 1, 2, 3, and 5 in the revised manuscript. Additionally, we have included the sample volume in the materials and methods section (Page 3, lines 103 and 107).